# Shallow Whole-Genome Sequencing from Plasma Identifies FGFR1 Amplified Breast Cancers and Predicts Overall Survival

**DOI:** 10.3390/cancers12061481

**Published:** 2020-06-06

**Authors:** Chantal Bourrier, Jean-Yves Pierga, Laura Xuereb, Hélène Salaun, Charlotte Proudhon, Michael R. Speicher, Jelena Belic, Ellen Heitzer, Brian Paul Lockhart, Nolwen Guigal-Stephan

**Affiliations:** 1Division of Biotechnology, Servier Research Institute, 125, Chemin de ronde, 78290 Croissy Sur-seine, France; chantal.bourrier@servier.com (C.B.); brian.lockhart@servier.com (B.P.L.); 2Department of Medical Oncology, Institut Curie, 26 rue d’Ulm, 75005 Paris, France; jean-yves.pierga@curie.fr (J.-Y.P.); salaunhelene@hotmail.fr (H.S.); 3Circulating Tumor Biomarkers Laboratory, Institut Curie, PSL Research University, INSERM CIC 1428, 26 rue d’Ulm, 75005 Paris, France; charlotte.proudhon@curie.fr; 4Université de Paris, 75005 Paris, France; 5Division of Methodology and Valorisation of Data, Servier Research and Development Institute, 50 rue carnot, 92150 Suresnes, France; laura.xuereb@servier.com; 6Institute of Human Genetics, Diagnostic and Research Center for Molecular BioMedicine, Medical University of Graz, Neue Stiftingtalstrasse 6, 8010 Graz, Austria; michael.speicher@medunigraz.at (M.R.S.); jelena.belic@cruk.cam.ac.uk (J.B.); ellen.heitzer@medunigraz.at (E.H.); 7BioTechMed-Graz, 8010 Graz, Austria; 8Christian Doppler Laboratory for Liquid Biopsies for Early Detection of Cancer, 8010 Graz, Austria

**Keywords:** clinical trials, *FGFR1*, liquid biopsy, breast Cancer, sWGS, ctDNA

## Abstract

**Background:** Focal amplification of fibroblast growth factor receptor 1 (*FGFR1*) defines a subgroup of breast cancers with poor prognosis and high risk of recurrence. We sought to demonstrate the potential of circulating cell-free DNA (cfDNA) analysis to evaluate *FGFR1* copy numbers from a cohort of 100 metastatic breast cancer (mBC) patients. **Methods:** Formalin-fixed paraffin-embedded (FFPE) tissue samples were screened for *FGFR1* amplification by FISH, and positive cases were confirmed with a microarray platform (Oncoscan^TM^). Subsequently, cfDNA was evaluated by two approaches, i.e., mFAST-SeqS and shallow whole-genome sequencing (sWGS), to estimate the circulating tumor DNA (ctDNA) allele fraction (AF) and to evaluate the *FGFR1* status. **Results:** Tissue-based analyses identified *FGFR1* amplifications in 20/100 tumors. All cases with a ctDNA AF above 3% (*n* = 12) showed concordance for *FGFR1* status between tissue and cfDNA. In one case, we were able to detect a high-level *FGFR1* amplification, although the ctDNA AF was below 1%. Furthermore, high levels of ctDNA indicated an association with unfavorable prognosis based on overall survival. **Conclusions:** Screening for *FGFR1* amplification in ctDNA might represent a viable strategy to identify patients eligible for treatment by FGFR inhibition, and mBC ctDNA levels might be used for the evaluation of prognosis in clinical drug trials.

## 1. Introduction

A plethora of genomic alterations, including mutations, amplifications, and gene fusions, have been described for the different members of the fibroblast growth factor receptor (FGFR) family in multiple cancer types [1]. In particular, amplification of *FGFR1* located at 8p11-12, resulting in overexpression, occurs in less than 10% across all breast cancer subtypes [2], in approximately 10–16% of luminal-type breast cancers [3,4,5,6], and 4% in triple-negative breast cancer [7]. *FGFR1* amplification and protein expression have been demonstrated as an unfavorable prognosis factor [5,8,9,10]. Several studies demonstrate that *FGFR1* amplifications also play an important role in resistance to endocrine therapy and possibly mediate this resistance mechanism [5]. In addition to ER resistance, *FGFR1* amplification has been demonstrated to confer broad resistance to PI3K and CDK4/6 inhibitors [11]. However, despite the clear rationale for targeting *FGFR1* in breast cancer with inhibitors, none have, to date, achieved an objective response, and this could be a consequence of inadequate patient selection and/or that *FGFR1* may not be the only oncogenic driver in the 8p11-12 amplicon.

In oncology clinical trials, one of the most important challenges is to ensure that the molecular status of the tumor accurately reflects the targeted population at the time of study inclusion. However, in clinical trials, it is sometimes difficult to obtain a fresh tissue biopsy at patient inclusion, especially in phase I trials and for exploratory research (for both ethical and/or medical health reasons). In addition, genomic anomalies present at relatively low to intermediate prevalence require extensive screening of patients. Consequently, molecular analyses, including mutation and somatic copy number alterations (SCNA) characterization, are typically performed on archival tissue, in most cases, acquired through a biopsy at diagnosis, which can significantly pre-date patient inclusion in clinical trials. Moreover, a single tissue biopsy (often a needle biopsy) is unlikely to reflect the global molecular, spatial, and temporal tumor dynamics with regard to both intra- and inter-tumoral heterogeneity, making clinical decisions—based on archival tissue biopsy—challenging.

Although less documented than SNV variation, several studies have also demonstrated both spatial and temporal evolution of gene amplification in cancer progression (in situ to invasive transition), from primary to metastatic tumors or under selective treatment pressure. The frequency of *FGFR1* amplification has been shown to be higher in invasive breast carcinomas and in the invasive components of tumors with both invasive and ductal carcinomas in situ (DCIS) components compared to pure DCIS [9]. De novo amplification of *FGFR1* has also been detected in a metastatic deposit of breast cancer (BC) patients not present in the primary tumor [12]. Likewise, *HER2* status might differ between primary and metastases tumors, and *HER2* amplification has been shown to be ‘acquired’ in about 2–3% of metastatic breast cancers, following neo-adjuvant chemotherapy treatment [13].

Consequently, based on the dynamics of cancer progression and significant intra-tumor heterogeneity, it is essential to evaluate the molecular profile at patient inclusion rather than relying exclusively on archival tissue. Detection of somatic alterations in cell-free DNA from plasma (cfDNA), which in cancer patients contains tumor-derived DNA fragments (ctDNA, circulating tumor DNA), is already in use in clinical practice and provides valuable information for patient management [14]. SCNA detection in plasma cfDNA by either digital droplet PCR or targeted next-generation sequencing (NGS) is much more challenging and not always cost-effective as mutation analysis. In this context, shallow whole-genome sequencing (sWGS) allows the identification of both genomic structural anomalies and ctDNA tumor fractions in plasma by measuring copy numbers from sequence read depth. This low-cost method has been successfully applied in several studies to monitor treatment response, to identify resistance mechanisms, or to estimate prognosis [15,16,17,18,19,20]. In this study, we used sWGS to assess the potential use of ctDNA as a surrogate for *FGFR1* amplification from metastatic breast cancer patients. In addition, we triaged patients based on their tumor content using two untargeted methods and correlated the basal ctDNA levels with patient survival.

## 2. Results

### 2.1. Characterization of the Cohort and Identification of FGFR1 Amplified Breast Cancer Cases

Detailed clinicopathological characteristics of the metastatic breast cancer (mBC) cohort (n = 100), covering different subtypes, are described in Table 1. A total of 88 tumors were estrogen receptor-positive (ER+) (88%), of which 60 tumors also demonstrated progesterone receptor positivity (ER+/PR+) (60%). PR+ alone also was observed in only one tumor. Eleven tumors were triple-negative. A total of 88 (92%) tumors were primary grade 2 and 3 tumors. The median follow-up was 72 months, and the median overall survival was 44 months. As previously reported, *FGFR1* [4] and triple-negative [21] tumors were associated with significantly worse survival (Kaplan–Meier log-rank *p* = 0.01; Appendix A) compared to hormone receptor-positive cases, with a shorter disease-free interval of less than 36 months (log-rank *p* = 0.035, Appendix A).

Using FISH, *FGFR1* amplification was identified in 20 cases (20%). Of those, 17 cases were classified as amplified and three cases as low amplified based on the criteria defined by Schildauss et al. [22] (Table 2 and Appendix A). The FGFR1/CEN8 ratio ranged from 1.1–8.4, with an average number of *FGFR1* gene signals per tumor cell of 4.3 ranging from 4.8–25.1. Among the 20 amplified *FGFR1* cases, 13 cases were ER+/PR+, 6 cases were ER+/PR−, and only one case was triple negative.

As a next step, whole-genome copy number analysis using the Oncoscan^TM^ technology was used as orthogonal confirmatory testing for *FGFR1* amplified cases. Out of the 20 FGFR1 amplification positive tumors identified with FISH, only 11 had sufficient amounts of DNA for the array analysis with no low amplified cases present (Table 3). With the exception of one case (N# 542827), Oncoscan confirmed the presence of *FGFR1* amplification, resulting in a concordance rate of 91%. Despite this high consistency in *FGFR1* amplification status between both technologies, the level of amplification only showed a weak correlation (*r = 0.399*) (Appendix A). At univariate analysis, *FGFR1* amplification was associated with shorter overall survival (OS) (median 32 months versus 54 months, log-rank *p* = 0.0018). During multivariate analysis, independent pejorative factors for OS were *FGFR1* amplification, triple-negative status, and a shorter disease-free interval (DFI).

### 2.2. SCNA Analysis in cfDNA and Evaluation of the Tumor Level

For ten of the initial 20 *FGFR1* amplified cases, corresponding plasma was available. In addition, 10 further cases without *FGFR1* amplification were randomly selected and analyzed as controls. cfDNA was initially analyzed using a modified fast aneuploidy screening test-sequencing system (mFAST-SeqS), which is basically a PCR with one primer pair that amplifies >20000 different LINE-1 sequences. The amplification product enables the calculation of a genome-wide z-score, which reflects aneuploidy and hence the level of ctDNA present in cfDNA [23]. For these 20 samples, the genome-wide z-score ranged from 1.1 to 31.5, with a median of 2.2. Only eight samples had a z-score greater than three, which was previously used as a threshold for further processing with sWGS (Figure 1 and Table 4). Since the resolution of mFAST-SeqS is limited to chromosome arms, and no LINE-1 sequences are located in the *FGFR1* amplified region, the amplification cannot be detected using mFAST-SeqS. However, high-level amplification can be detected with sWGS down to 1% [23,24]. Therefore, we ran the same samples with sWGS irrespective of the mFAST-SeqS z-score (for detailed copy number profiles, see Appendix A). For copy number calling, we used our previously published plasma-Seq [15] approach, while the ichorCNA algorithm [13] was used to calculate the tumor fraction. A good correlation (R^2^ = 0.81) between z-score and ichorCNA estimates was observed (Figure 2), essentially driven by the high content tumor DNA cases.

It is well established that SCNA analysis from cfDNA requires a minimal ctDNA AF. For the ichorCNA algorithm, a lower limit of 0.03 tumor fraction has been reported to detect the presence of tumors with high sensitivity (0.95) and specificity (0.91) [17]. From the 20 samples, for which we conducted plasma analyses, 12 had a ctDNA AF above 3%, 7 of them had an *FGFR1* amplification, whereas the other 5 cases had no copy number alteration of the *FGFR1* region. In all of these 12 cases, the presence or absence of the *FGFR1* amplification was correctly identified.

According to ichorCNA, the other three cases with *FGFR1* amplifications had low ctDNA AFs of 0.35% (529014), 1.2% (542827), or 0.86% (562315). However, despite the low ctDNA AF of 0.86%, the *FGFR1* amplification was correctly identified in case 562315. Not surprisingly, the tissue evaluation of this case showed by far the highest number of signals, i.e., 25.1 (FISH) or 24 (Oncoscan) (Figure 3). This is a case in point that accurate resolution limits for amplifications are hard to establish as they critically depend on the level of amplification. In contrast, in case 529014, the ctDNA AF of 0.35% and about seven *FGFR1* copies/nucleus (FISH and Oncoscan) were apparently insufficient for the detection of the amplification. In the third case 542827, the results between FISH and Oncoscan had already been inconsistent so that the presence of the amplification was questionable (Figure 3). When we correlated the log2ratio of the 1Mb window harboring the *FGFR1* gene, we observed a low correlation with the tumor fraction assessed with ichorCNA (iTF) for the amplified cases, and, as expected, no correlation for non-amplified cases of the respective region was balanced (or maybe only affected by a large gain) regardless of the tumor fraction (Appendix A). Additionally, we calculated the (hypothetical) absolute copy numbers of *FGFR1* based on the log2ratio and tumor fraction estimation from ctDNA. Except for one outlier, these hypothetical copy numbers were highly correlated to Oncoscan copy number but not to FISH (Appendix A).

Since the detection of *FGFR1* amplification in plasma critically depends on both the absolute copy number of the gene (amplification amplitude) and then tumor fractions, a generalized threshold for the detection could not be established.

### 2.3. cfDNA Tumor Fraction Could Predict Survival

Previous reports using targeted methodologies (dPCR, ddPCR, and targeted-NGS) [25] and a recent study with mFAST-SeqS [26] have demonstrated that higher tumor levels in plasma are indicative of a worse prognosis in breast cancer. In this report, we wished to demonstrate the suitability of evaluating tumor fraction by ichorCNA using ULP-WGS in a metastatic breast cancer cohort. First, we used the median genome-wide z-score for stratification. Indeed, we found that z-scores above the median of 2.2 were associated with significantly decreased overall survival (OS) (*p*-value for the log-rank test; *p* = 0.009), yet with slightly overlapping CI (Figure 4a). Likewise, splitting patients, based on the median tumor fraction established with ichorCNA (ichorCNA median 4.6), led to a similar separation of the Kaplan–Meier curves (*p*-value for the log-rank test; *p* < 0.001) (Figure 4b). On the contrary, the total cfDNA quantity estimated by Qubit (based on median cut-off) was not found relevant to discriminate OS (Figure 4c).

## 3. Discussion

Despite the clear rationale for targeting *FGFR1* in BC and multiple therapeutic strategies based on both selective and non-selective *FGFR1* inhibitors, none have, to date, achieved an objective response. The moderate signs of efficacy could be the cumulative effect of ineffective compounds, inadequate patient selection, or the lack of oncogenic potential of *FGFR1*.

In this study, we successfully demonstrated the feasibility of sWGS (plasma-Seq) for detecting genome-wide structural genomic anomalies, including *FGFR1* gene amplification, as well as estimating tumor content in the plasma of mBC patients. Despite the small sample size, good concordance was observed between *FGFR1* amplification in plasma assessed from sWGS data and the tissue-based analysis (FISH and Onscoscan^TM^). Therefore, the method has the potential to directly screen and detect *FGFR1* amplification in a non-invasive and specific manner in plasma samples in patients with advanced metastatic disease, demonstrating significant tumor burden.

Amplification calling in plasma, compared to mutation detection, is much more challenging, where the ratio between the ctDNA and normal cfDNA determines the capacity of defining amplification status. Indeed, several targeted NGS methods based on amplicon-based multiplexed PCR and hybrid-capture have been described for detecting specific SCNAs [27], but high coverage and in-depth sequencing requirements, as well as lack of cost-effectiveness, have somewhat limited their widespread clinical implementation. In addition, cost-effective methods for detecting SCNAs in cfDNA based on ddPCR have so far only been described for *ERBB2 (HER2)* [28] and *FGFR2* [29]. Although sWGS can specifically detect both focal and gross copy number changes, its major limitation is the lack of sensitivity to detect low copy number gain/loss and smaller genomic changes due to the low tumor DNA component in early-stage cancer. In addition, another limitation in using sWGS in estimating plasma copy numbers is that only relative copy number changes are measured, which neither allows establishing the ploidy level nor the exact prediction of the absolute copy number due to the dilution effects with DNA from normal cells [15]. Despite these shortcomings—and although the evaluation of cost-effectiveness and duration were not within the scope of the study—sWGS (and mFAST-SeqS) still represented a pan-cancer, low-cost, and rapid screening approach to specifically detect global genomic structural anomalies in cfDNA to enable the identification of actionable therapy and also monitoring in advanced cancer patients. The turnaround time of these methods is less than two days at costs of less than 100EUR. Further validation in a larger cohort will be required to reinforce the potential of this method for screening the *FGFR1* amplification.

In addition, and despite the small cohort size, both untargeted sequencing methods—mFAST-SeqS and sWGS—demonstrated similar results in predicting survival based on tumor content in metastatic breast cancer (mBC) patients. Although the estimation of tumor fraction in plasma with each method (mFAST-SeqS and ichorCNA) demonstrated high correlation, ichorCNA might be a more accurate representation of the tumor fraction and, therefore, enable a better stratification (*p <* 0.001 versus *p =* 0.009). mFAST-SeqS is based on *LINE-1* read counts, and the z-score depends on the overall amount of copy number alterations, as well as the amplitude of copy number alterations, while ichorCNA interrogates the entire genome. The specificity of these methods is reinforced by the fact that measuring global cfDNA levels in the plasma of the same patients is not predictive of survival. Both methods have been previously proven to be of prognostic value in breast cancer [17,26]. The mFAST-SeqS-based assessment has also been used in the prostate [24,30] and lung cancer [31]. Highly concordant copy number profiles of mFAST-SeqS and plasma-Seq were observed in patients’ plasma samples with z-score >3. However, in contrast to sWGS, mFAST-SeqS did not allow to call for *FGFR1* amplification, as there was no *LINE*-1 sequence near the *FGFR1* gene.

Considering the entire cohort of 100 patients, we confirmed previous reports that *FGFR1* amplification based on tissue analyses is associated with significantly worse survival [5,6,9,32,33]. Unfortunately, due to the limited number of patients with available plasma, we were unable to perform a similar analysis in plasma with sufficient statistical power.

## 4. Materials and Methods

### 4.1. Clinicopathologic Data and Sample Collection

Metastatic breast tumors and blood samples were selected from 100 female patients at the Institute Curie, Paris, based on the availability of both tissue and plasma samples (Table 1). Blood sampling was performed at diagnosis of metastasis or before the beginning of a new line of treatment for metastatic disease. In accordance with the French regulation, written consent was obtained from all patients/guardians on specimens. This research has been approved by the institutional review board of the Institut Curie on 11 June 2015. The patients were treated for stage IV disease at the Institute Curie, Paris, from 4/2006 to 4/2018, with a median follow-up of 6 years and a median overall survival of 44 months. All patients received systemic treatment (endocrine and/or chemotherapy) according to tumor characteristics and standard of care at the time of inclusion. The median age of patients in the cohort was 60y (range 28–83). The details of the clinical and histopathological data care are summarized in Table 1. Tissue samples were fixed in formalin, embedded in paraffin (FFPE), and stored under standard conditions. Blood was drawn into standard EDTA tubes. Plasma was extracted by centrifugation at 2000 rpm for 10 min, followed by careful aliquotting and freezing at −80 °C within 1–24 h after collection.

### 4.2. Fluorescence In Situ Hybridization (FISH)

FISH was performed on 5 µm tumor sections. We used a commercially available and standardized probe for the detection of FGFR1 (Zytovision^TM^), and hybridization was performed according to the manufacturer’s instructions. FGFR1 was assessed at a central facility (Zytovision^TM^) on available archived metastatic biopsy. Where possible, we scored 50 cells per sample for hybridization patterns. FGFR1 amplification was defined, based on published criteria [22]; notably, the FGFR1/CEN8 ratio was ≥2.0, or the average number of FGFR1 signals per tumor cell nucleus was ≥6. Additional evaluation criteria (FGFR1 low amplification) were considered if the percent of tumor cells containing >5 gene signals was >50% and/or % of tumor cells containing >15 gene signals or large clusters was >10%.

### 4.3. Genomic Oncoscan Profile

DNA was isolated from FFPE tumor samples using the GeneRead DNA FFPE Extraction Kit (Qiagen, Hilden, Germany), following the manufacturer’s guidelines. The genome-wide copy number profile, including regions with loss of heterozygosity (LOH), was established using Oncoscan^TM^ arrays, according to the supplier’s recommendation (Affymetrix-Thermofisher Scientific, Inc., Santa Clara, CA, USA, ONCOSCAN CNV FFPE ASSAY, ref. 902293) and as described previously [34,35]. Data analysis was performed using Chromosome Analysis Suite Software from Affymetrix (CHAS), version 3.0.0.42, with annotations na33.1 (hg19). QC metrics were generated and gave information about the data quality and the level of chromosomal aberrations. Comprehensive Pipeline for Analyzing and Visualizing Array-Based CGH Data R Package was used for whole-genome representation [36] (rCGH: version 1.14.0).

### 4.4. Cell-Free DNA Extraction and Quantification

cfDNA was extracted from 1 to 4 mL of plasma using Maxwell RSC ccfDNA Plasma Kit (RSC; Promega, Leiden, the Netherlands), according to the manufacturers’ recommendation. The RSC kit is a magnetic beads-based method, fully automated, including a lysis step before DNA binding to magnetic beads. cfDNA concentration was measured by Qubit Fluorometric Assay (Thermo Fisher Scientific, Aalst, Belgium) with dsDNA HS (High Sensitivity) Assay Kit. The total cfDNA quantity per mL of plasma was calculated.

### 4.5. mFAST-SeqS

To stratify patients based on their tumor levels, mFAST-SeqS was used, as previously described [30,37]. Briefly, LINE-1 (L1) amplicon libraries were prepared from 5–10 µL of plasma DNA corresponding to 0.1–5 ng total DNA using Phusion Hot Start II Polymerase in 5 PCR cycles with target-specific L1 primers. The 10 µL of the purified (AMPure Beads, Beckman Coulter, Brea, CA, USA) PCR products were used for the addition of Illumina-specific sequencing adaptors and indices. L1 amplicon libraries were sequenced on an Illumina MiSeq, generating 150 bp single reads aiming for at least 100,000 reads per sample. Sequence reads were aligned to the hg19 genome, and reads with a mapping quality >15 were counted per chromosome arm with an in-house script. Normalized read counts were compared to a control population (*n* = 35) using z-score statistics by subtracting the mean and dividing by the SD to assess over- and under-representation of LINE-1-sequences. The short arms of acrocentric chromosomes were omitted from the analysis. Finally, all chromosome arm-specific z-scores were squared and summed up, resulting in a genome-wide z-score, which acts as a surrogate for tumor fraction [30].

### 4.6. Shallow Whole-Genome Sequencing sWGS (PLASMA-SEQ)

For genome-wide copy number profiling, sWGS (plasma-Seq) was performed, as previously described [14]. Briefly, shotgun libraries were prepared from 5–10 ng plasma DNA using the TruSeq DNA Nano Sample Preparation Kit (Illumina, Inc., San Diego, CA, USA). Due to the fragmentation of plasma DNA and the low input amounts, we omitted the fragmentation step and increased the number of PCR cycles to 20. Libraries were quality checked on an Agilent Bioanalyzer (Agilent, Santa Clara, CA, USA) and quantified using qPCR. Finally, libraries were sequenced on an Illumina MiSeq or NextSeq with an average of 6.43 mio. reads (range 4.7–8.5 mio.) per sample. Large copy number changes, as well as focal events, were established using our optimized in-house plasma-Seq script [14,24]. *FGFR1* copy number status was evaluated based on the previously published criteria for focal amplification calling. Among other genes, such as *AR ERBB2*, *MET*, and *MYC*, *FGFR1* amplification was already validated using quantitative real-time [22,38]. Additionally, data were analyzed with the previously published ichorCNA algorithm to calculate tumor fractions (iTF) from sWGS data [17]. The hypothetical absolute copy number (ACN) of the 1 Mb window harboring the FGFR1 gene was calculated by formula (2): ACN = (2∗(2^log2Ratio + iTF – 1))/iTF [39].

### 4.7. Statistical Analysis

Descriptive statistics were used in order to analyses the relationship between variables. Kaplan–Meier analysis was used to estimate overall survival. To stratify patients, the median values of the genome-wide z-score assessed with mFAST-SeqS and the tumor fraction calculated from ichorCNA were used. Survival was defined as the time from the date of measurement for ctDNA data to the date of death, the date of last news, or to 26 January 2018 (right censoring). The Kaplan–Meier curves and their 95% confidence intervals, as well as the survival medians (in months), were presented by the group. All statistical analyses were performed using SAS^®^/PC Software version 9.2.

For all the correlations presented, Pearson correlation coefficients were performed with GraphPad Prism.

## 5. Conclusions

In summary, we demonstrated the feasibility of using sWGS as an assay to detect clinically informative SCNA, such as *FGFR1* amplification, in plasma of mBC patients. In the cases where access to fresh biopsies is not possible or that only archived samples are available, sWGS could be used for screening or re-evaluating the amplification status of the patients. In addition, both mFAST-SeqS and sWGS confirmed their potential in a broad-based application for the estimation of tumor fraction and, thus, tumor monitoring in advanced cancer patients.

## Figures and Tables

**Figure 1 cancers-12-01481-f001:**
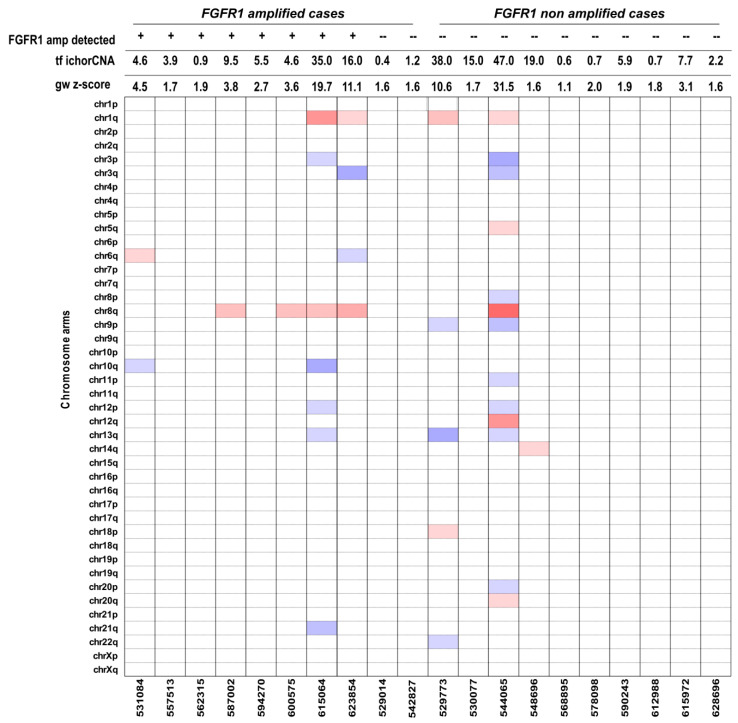
mFAST-SeqS (modified fast aneuploidy screening test-sequencing system) LINE-1 profiles for *FGFR1* amplified (*n* = 10) and non-amplified cases (*n* = 10). Heat-map of selected plasma samples with different values of genome-wide z-scores. Blue bars indicate chromosome-specific Z-scores < 5, and red bars indicate chromosome-specific z-scores >5. Amongst the 20 samples analyzed with mFAST-SeqS, the z-score ranged from 1.1 to 31.5 with a median of 2.2m with only 4 samples showing a Z-score > 5.

**Figure 2 cancers-12-01481-f002:**
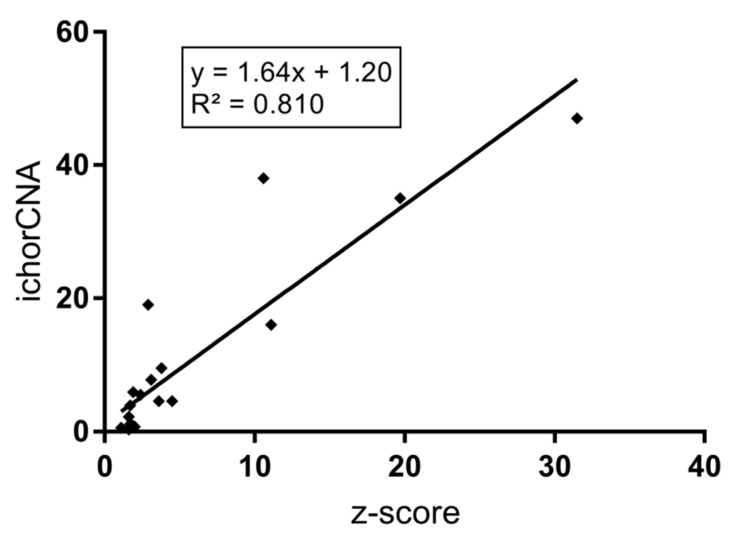
Correlation (R^2^ of 0.81) between mFAST-SeqS z-score and ichorCNA.

**Figure 3 cancers-12-01481-f003:**
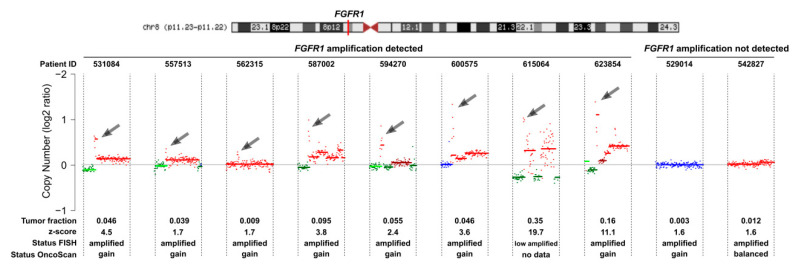
Copy number status of chromosome 8 of cell-free DNA (cfDNA) samples established with shallow whole-genome sequencing (sWGS). The log-2 ratios of chromosomes 8 of plasma samples from 10 metastatic breast cancer patients with *FGFR1* amplification in their tumors are plotted. In 8 of those, the *FGFR1* amplification could also be detected in plasma (right panel), whereas in 2 cases, the amplification could not be detected most likely due to a low tumor content. The gain of chromosomal regions is indicated in red, losses are shown in green, and balanced regions are shown in blue. Tumor fraction estimated from ichorCNA, z-score, FISH, and Oncoscan status is indicated.

**Figure 4 cancers-12-01481-f004:**
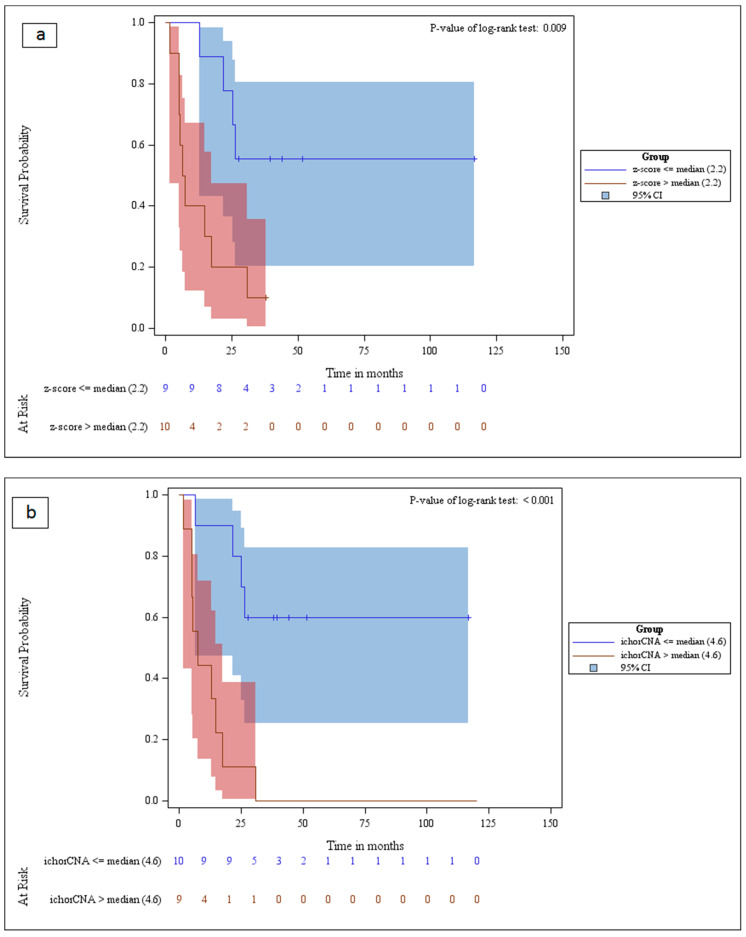
Kaplan–Meier analysis curves for the overall survival of metastatic breast cancer patients. (**a**) Overall survival of patients stratified by the median mFAST-SeqS z-score, (**b**) Overall survival of patients stratified by the median tumor fraction calculated with ichorCNA cut-off, (**c**) Overall survival of patients stratified by the median cfDNA quantity established by Qubit.

**Table 1 cancers-12-01481-t001:** Clinicopathological characteristics of metastatic breast cancer cohort. * 4 missing data ** 1 missing data. TNBC: triple-negative breast cancer, ER: estrogen receptor, PR: progesterone receptor, DFI: disease-free interval (between primary and distant metastasis diagnosis), IDC: invasive ductal carcinoma, FGFR1: fibroblast growth factor receptor 1.

Caption	FGFR1 Amplified	FGFR1 Amplified	Total
	*N* = 20	*N* = 80	*N* = 100
AGE (years) median	56 (28–78)	60 (32–83)	60 (28–83)
<50 year	4/20	25/80	29
DFI (months) median	35 (0–183)	31 (0–227)	33 (0–227)
IDC	18 (90%)	71 (89%)	89
Lobular	2 (10%)	9 (11%)	11
Primary tumor grade *	-	-	-
1	0	8	8
2	9	32	41
3	11	36	47
ER+/−	19/1	69/10	88/11
PR+/−	13/7	47/27	60/34
TNBC **	1	10 (out of 79)	11 (out of 99)
Visceral metastasis y/n	15/5	40/40	55/45
Metastatic sites >2 y/n	10/10	23/57	33/67

**Table 2 cancers-12-01481-t002:** *FGFR1* amplified cases based on FISH testing of a metastatic breast cancer cohort (*n* = 100). * *FGFR1* amplification was based on the criteria of Schildhaus et al., 2012. Notably: FGFR1/CEN8 ratio was ≥2.0, or the average number of *FGFR1* signals per tumor cell nucleus was ≥6. Additional evaluation criteria (*FGFR1* low amplification) were considered if the % of tumor cells containing >5 gene signals was >50% and/or % of tumor cells containing >15 gene signals or large clusters was >10%.

Caption	Evaluation Criteria * *FGFR1* was Amplified, If *FGFR1/CEN8* Ratio was ≥2.0 and/or Average Number of *FGFR1* Signals/Tumor Cell was ≥6.0				Additional Evaluation Criteria (Low Amplification) *FGFR1* was Amplified, If the Percentage of Tumor Cells Containing ≥5 Gene Signals was ≥50% and/or Percentage of Tumor Cells Containing ≥15 Gene Signals or Large Clusters was ≥10%		
Case No. (Patient ID)	Target Gene/Centromere Ratio	Average Number of Target Gene Signals/Nucleus	Average Number of Centromere Signals/Nucleus	*FGFR1* Status	Percentage of Tumor Cells Containing ≥5 Gene Signals (Quotient)	Percentage of Tumor Cells Containing ≥15 Gene Signals or Large Clusters(Quotient)	*FGFR1* Status
446894	2.3	6.3	2.8	amplified	0.8	0	amplified
525973	1.9	7.7	4.1	amplified	0.9	0.1	amplified
529014	3.4	7.2	2.1	amplified	0.9	0	amplified
531084	2.5	6.3	2.5	amplified	0.8	0	amplified
537264	1.1	7.5	7.1	amplified	0.9	0.1	amplified
542827	4.5	11.6	2.6	amplified	1.0	0.3	amplified
548602	6.7	16.2	2.4	amplified	1.0	0.6	amplified
550066	1.9	5.9	3.1	balanced	0.8	0	amplified
557513	5.5	10.8	2.0	amplified	1.0	0.3	amplified
562315	8.4	25.1	3.0	amplified	1.0	0.9	amplified
587002	3.4	15.6	4.6	amplified	1.0	0.7	amplified
594270	2.8	11.6	4.2	amplified	1.0	0.3	amplified
595763	2.5	10.5	4.2	amplified	1.0	0.1	amplified
599471	5.4	9.8	1.8	amplified	1.0	0.1	amplified
600575	6.2	12.8	2.0	amplified	1.0	0.4	amplified
615064	1.9	4.8	2.6	balanced	0.6	0	amplified
621823	5.6	12.7	2.2	amplified	1.0	0.4	amplified
623854	4.6	7.8	1.7	amplified	0.7	0.1	amplified
630510	4.2	5.3	1.3	amplified	0.6	0	amplified
631774	1.9	5.1	2.7	balanced	0.6	0	amplified

**Table 3 cancers-12-01481-t003:** *FGFR1* amplification status concordance between FISH and Oncoscan testing.

Case No. (Patient ID)	*FGFR1* Status Oncoscan	*FGFR1* Copy Number Oncoscan	Concordance with FISH	Biopsies-FISH	Copy Number: Average Number of Target Gene Signals/Nucleus
529014	Gain	3	Yes	Amplified	7.2
531084	Gain	4	Yes	Amplified	6.3
542827	No gain	2	No	Amplified	11.6
548602	Gain	5	Yes	Amplified	16.2
557513	Gain	8	Yes	Amplified	10.8
562315	Gain	24	Yes	Amplified	25.1
587002	Gain	5	Yes	Amplified	15.6
594270	Gain	16	Yes	Amplified	11.6
600575	Gain	7	Yes	Amplified	12.8
623854	Gain	15	Yes	Amplified	7.8
630510	Gain	5	Yes	Amplified	5.3

**Table 4 cancers-12-01481-t004:** Amplification calling based on FISH, Oncoscan testing, and plasma-Seq data.

Case No. (Patient ID)	Biopsies-FISH	Oncoscan-Curie	mFAST-SeqS Z-Score	ichorCNA Tumor Fraction (%)	*FGFR1* Amplification Calling Based on Plasma-Seq Data
529014	Amplified	Gain	1.6	0.35	-
529773	Balanced	No Data	10.6	38	-
530077	Balanced	No Data	1.7	1.15	-
531084	Amplified	Gain	4.5	4.6	+
542827	Amplified	No gain	1.6	1.2	-
544065	Balanced	No Data	31.5	47	-
548696	Balanced	No Data	2.9	19	-
557513	Amplified	Gain	1.7	3.9	-
562315	Amplified	Gain	1.9	0.86	+
568895	Balanced	No Data	1.1	0.6	-
578098	Balanced	No Data	2	0.73	-
587002	Amplified	Gain	3.8	9.5	+
590243	Balanced	No Data	1.9	5.9	-
594270	Amplified	Gain	2.4	5.5	+
600575	Amplified	Gain	3.6	4.6	+
612988	Balanced	No Data	1.8	0.7	-
615064	Amplified low	No Data	19.7	35	+
615972	Balanced	No Data	3.1	7.77	-
623854	Amplified	Gain	11.1	16	+
628696	Balanced	No Data	1.6	2.2	-

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
