# Peer review of "Shallow Whole-Genome Sequencing from Plasma Identifies FGFR1 Amplified Breast Cancers and Predicts Overall Survival"

_cancers, 2020, doi:10.3390/cancers12061481_

Round 1

Reviewer 1 Report

The manuscript of BOURRIER et. al. presents cfDNA that was evaluated by two approaches, i.e. mFAST-SeqS and shallow whole‐genome sequencing (sWGS), to estimate the circulating tumor DNA allele fraction and to evaluate the FGFR1 status. The high levels of ctDNA have indicated an association with a bad prognosis based on overall survival. The work is very interesting and the approach is promising. I would suggest to address the following questions:

  1. Please present a graph of ctDNA concentration vs. a number of amplification in the FGFR1 gene. Does it have any correlation or not? How could you explain it?
  2. Please use RPKM or FPKM to evaluate the "pieces" of FGFR1 in cfDNA that have been identified for each person vs. their corresponding amplification rate. Is there any correlation? Could you suggest a measure for the FGFR1 amplification using cfDNA sequencing methods, RPKM, FPKM or any other? Add an explanation and figures in the manuscript, it might be very useful for clinicians and researchers in the field.

Reviewer 2 Report

In the present manuscript, the authors attempted to evaluate the feasibility of a plasma-based NGS method for detection of FGFR1 amplification and to assess the prognostic role of ctDNA level in metastatic breast cancer. Although conceptually intriguing, this study is flawed by a number of shortcomings which limits very much its potential value:

  • The sample size of patients with the availability of plasma among those with tissue amplified FGFR1 cases (n=10) is too small to build on a research paper since the inherent risks of biases are too high. A scientific contribution in the form of a preliminary report or a letter to Editor would be preferable.
  • The study population is far from being clinically well-annotated: did the patients receive any treatment? If so, what type of treatment they received? What is the burden of disease for each patient? What are the sites of disease? These are established factors that may influence the amount of ctDNA and have an impact on patients' outcome.
  • No information whatsoever is provided as to the timing and the number of blood sample collection. As a consequence, it is almost impossible to discuss their clinical reliability and usefulness.
  • The methodology applied to the profiling of plasma samples is quite investigational and a validation process using better-standardized techniques such as droplet digital PCR would be needed.
  • Although the ctDNA amount has shown to impact on patients survival, no prognostic role for plasma amplified FGFR1 has been demonstrated. 

Reviewer 3 Report

Dear Authors,
Thank you for the interesting and well-designed paper. Here there are my comments/questions to this hardworking and well-done study.

The authors have identified FGFR1-amplification in 20 % patients by FISH and Oncoscan. The unique advantage of Oncoscan is of course, the possibility of performing whole-genome copy number analysis from FFPE samples, which can be applied in archival tissue, if new biopsy is not feasible. And there was found an excellent correlation of 91%. However, the incidence of FGFR1-amplification may be higher. Have you investigated the whole cohort of 100 patients with any NGS panel in order to identify the FGFR1 amplification? There may be cases positive only for FISH, positive only for NGS and positive for both (as e.g. for MET-amplification).

You have investigated the correlation between the level of amplification and survival for FISH and Oncoscan.
As we know e.g. for ALK-rearranged NSCLC, ALK-FISH with 50 cell count (> 50% positive nuclei) versus 100 cell count (< 50% positive nuclei) is correlated with better PFS while treated with ALK-TKI, can you define any potential cut-off value of FGFR1-FISH regarding both % of cells containing > 5 and > 15 gene signals?

Out of 20 FGFR1-amplified patients only for 10 patients plasma was available. Under condition that there was a little group, do you think that AF above 3% can be a borderline for the dependability of the method?
Figure 4.b showing the curves calculated with ichorCNA cut off (≤ 4.6 and > 4.6) has demonstrated even more convincing p-value of log-rank test < 0.001. Can you comment something more in discussion - despite of concordance of these curves in the 3 methods presented in figure 4 – the differences of them and which features they describe, as they reflect different aspects of perception of FGFR1-amplification.

Finally, and in terms of its practical application as a screening method, what was your TAT (turnaround time) for mFAST-SeqS and sWGS?

Additionally there are some small typographical errors:
line 49, 67 and 69 - FGFR1 should be in italics
line 64 - extension of abbreviation should precede SNV, which should be in brackets as used the first time
line 70 and 71 - HER2 should be in italics
line 106 - FGFR1 should be in italics and amplificated instead of amplification
line 99, 112, 114, 125, 164, 196, 199 x 2, 200, 202, 204, 206, 208, 226, 236, 252, 253, 255x 2, 245, 257, 298, 313, 320, 323, 324, 325, 326 x 2, 327, 328 (and table/legend of nr 2, 3, 4, figures 1 and 3) – FGFR1 should be in italics
line 113 and 199 - double space
line 336 - too many dots after E.H. initials
line 404 - reference nr 20 is missing

Reviewer 4 Report

In general, this is a well written manuscript in proper English. No extensive revision of the English language is required. Nevertheless, there are a few minor typos, that I will address, next to some issues I came across reading this manuscript.

Page 2, LN 49: it can account for up to 23% of the luminal subtypes of breast cancer (Int. J. Mol. Sci 2020; 21, 2011). FGFR1 amplification is also seen in triple negative breast cancer (TNBC), especially in basal like and mesenchymal like subtypes. As this is the case, I would also suggest to mention overall prevalence of FGFR1 amplification in breast cancer, regardless of intrinsic subtypes. In page 2, LN 95, you finally mention the presence of FGFR1, according to a not aforementioned, new reference. As this study includes metastatic TNBC, the introduction lacks the rationale and aims for testing FGFR1 amplification in this specific subtype. I would therefore suggest to add this in the manuscript.

Page 2, LN 50-51: it is unwise to state that the presence of an alteration supports directly the oncogenic potential of this alteration, as multiple actors will need to count towards this oncogenic potential. This is usual the case in alterations of growth factor receptors. This might also be the reason why no objective response is achieved with the administration of FGFR(1)-inhibitors.

Page 2, paragraph 2: I do not fully agree that obtaining a fresh biopsy at patient inclusion in clinical trials should impede patient recruitment. If obtaining a fresh biopsy is required for a clinical trial, this should have been approved by an ethical committee in the first place. Medical health reasons might be a reason for inhibiting inclusion. Yet, in breast cancer, it is more likely due to unreachable metastatic locations than due to adverse effects of taking a fresh biopsy. I do agree that a biopsy should correlate with the state of disease to be evaluated. Therefore, archived biopsies are usually not the most optimal tissue samples to be used in a clinical trial. This and tumor heterogeneity (which is certainly seen in breast cancer) do certainly count towards the rationale of using cell free DNA from plasma.

Page 2, section “Introduction”: the introduction mentions the presence of FGFR1 amplifications in luminal-type breast cancers, but the introduction does not mention the most important role of FGFR1 amplifications in this intrinsic subtypes. It does not only demonstrate a worse prognosis or a potential target of therapy, but FGFR1 amplifications also play in important role in resistance to endocrine therapy and possibly mediate this resistance mechanism.

Page 2, LN 95: see also Page 2, LN 49.

Page 4, Table 1: although the ratio IDC/lobular is correct, it is hard to read. I suggest to create a few extra rows for separating both categories and providing proportional values in percentages. The caption mentions the word “IntervalPg”, which might be a typo. Furthermore, as FGFR1 amplifications plays an important role in resistance to endocrine therapy, It would be best to provide how many of the FGFR1 amplified and non-amplified breast cancer patients have been administered endocrine therapy.

Page 9, Section “Discussion”: the first paragraph doesn’t add any value to the manuscript. Most of the sentences are reused form the first paragraph of the introduction.

Page 9, LN 211: “Nevertheless” implies a soft contradiction to the previous sentence, but the next sentence rather agrees. It could be changed to “Therefore”

Page 10, LN227 and Page 12, section “Conclusion”: It is stated that sWGS is low-cost and can be used as rapid screening. Although experience-based, this might be true, this conclusion cannot be made from this study, as cost-effectiveness and duration were not a part of this study design. There are also no references to studies comparing sWGS to its alternatives. I would either drop this suggestion or refer to these claims.

Round 2

Reviewer 2 Report

The authors have addressed all the raised comments and/or suggestions in a satisfying fashion.

Reviewer 4 Report

Only one small error detected: p11, 328: ....using quantitative real-time....I think "PCR" was omitted.